# Extended-Reach Drilling (ERD)—The Main Problems and Current Achievements

Karim El Sabeh, Nediljka Gaurina-Međimurec *, Petar Mijić, Igor Medved and Borivoje Pašić

Faculty of Mining, Geology and Petroleum Engineering, University of Zagreb, 10000 Zagreb, Croatia
* Correspondence: nediljka.gaurina-medjimurec@rgn.unizg.hr; Tel.: +385-1-5535-825

**Abstract:** With the development of different segments within the drilling technology in the last three decades, well drilling has become possible in harsh downhole conditions. The vertical well provides access to oil and gas reserves located at a certain depth directly below the wellsite, and a large number of vertical wells are required for the exploitation of hydrocarbons from spatially expanded deposits. However, the borehole can deviate from the vertical well, which means that the target zone can be reached by a horizontal directional well. With this type of well, especially in the case of drilling an extended-reach well (ERW), the length of the wellbore in contact with the reservoir and/or several separate reservoirs is significantly increased, therefore, it is a much better option for the later production phase. Unfortunately, the application of extended-reach drilling (ERD technology), with all of its advantages, can cause different drilling problems mostly related to the increased torque, drag, hole cleaning and equivalent circulation density (ECD), as well as to an increase in the well price. Overcoming these problems requires continuous operational change to enable operators to address downhole challenges. Today, the longest well reaches 15,240 m (50,000 ft), which raises the question of the technological and economic feasibility of this type of drilling project, especially with the lower oil price on the energy market. This paper provides a comprehensive overview of extended-reach drilling technology, discusses the main problems and analyzes current achievements.

**Keywords:** well design; extended-reach drilling (ERD); torque; drag; hole cleaning

## 1. Introduction

From its beginning, the oil and gas (O&G) industry, especially exploration and production (E&P) activities, have passed through significant changes in every way. This remarkable technical and technological development is best seen in the drilling and completion of oil and gas wells all around the world. In the beginning, the drilled well was shallow and vertical and wellbore deviation was undesirable, which were consequences of the underdevelopment of drilling technology as well as a lack of proper engineering knowledge. In other words, the main goal was drilling off the straight wellbore directly from the surface location to a certain underground point, mostly located directly beneath the future wellhead. With time, driven by an increasing need for energy, oil and gas companies discovered new reserves of oil and gas, often at a greater depth and inaccessible places such as the deep sea, populated areas or environmentally sensitive areas. This was exactly the main initiator of the change in drilling technology in the last few decades, which had just one aim: to enable a technological and cost-effective development of the new reservoirs.

Directional drilling is certainly one of the most important technical and technological improvements in drilling technology, enabling us to reach oil and gas reserves that would be unreachable with simple vertical wells. According to the API (American Petroleum Institute), controlled directional drilling can be defined as *"The art and science involving the intentional deflection of a wellbore in a specific direction in order to reach a predetermined objective below the surface of the earth, and today, it is much more science than art"* [1].

Directional drilling technology, and especially extended reach technology, is continuously evolving, and, in the last twenty years, the world is witnessing world record

breaking in terms of drilling the longest wells. Today, engineers are capable of designing and successfully drilling wells deeper (or rather longer) than 15,240 m (50,000 ft), onshore or offshore. Although extended-reach technology provides numerous benefits over conventional vertical or directional drilling technology, it simultaneously presents a kind of engineered challenge. Hole cleaning problems, an increase in drag and torque an an increased equivalent circulation density are just some of the problems encountered by engineers and, unfortunately, with an increase in wellbore length, the problems become more challenging and difficult to overcome.

Today, reservoirs can be developed by wells with a different well construction (Figure 1), even on the same field, and the construction of the particular well depends on numerous factors, such as the reservoir depth and type, local lithology, proven reserves, expected problems, costs, etc.

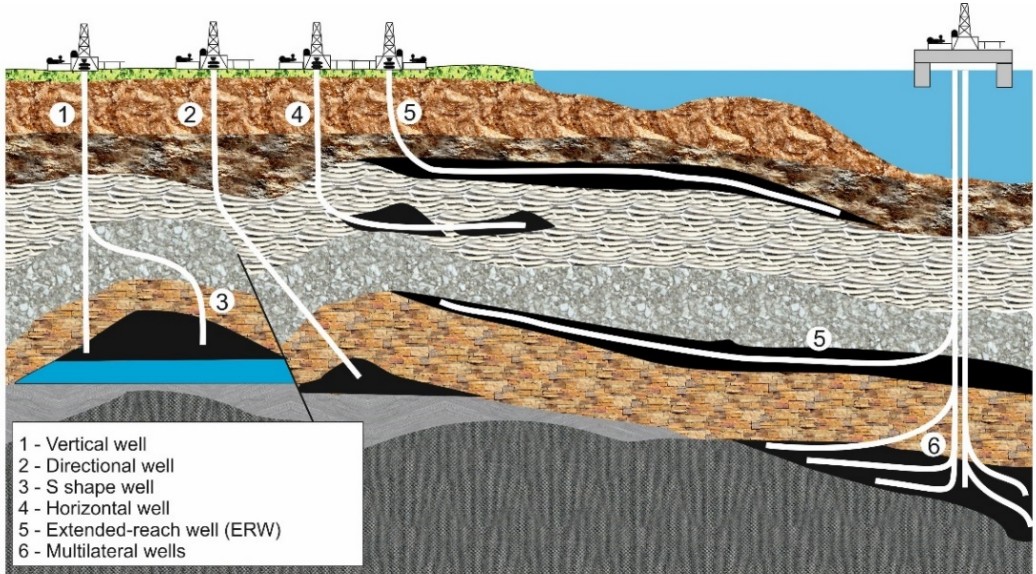

**Figure 1.** Different well types (Reprinted with permission from Ref. [2]. 2023, Croatian Academy of Engineering).

Extended-reach drilling (ERD) technology is essentially an advanced form of directional drilling or horizontal drilling technology. Extended-reach drilling employs both directional and horizontal drilling techniques. It has the ability to achieve horizontal well departures and total vertical depth-to-horizontal distance ratios well beyond conventional directional drilling [3]. An extended-reach well is defined as a well where the ratio between the horizontal reach (departure) and true vertical depth (TVD) is larger than 2, or where the horizontal displacement is greater than 20,000 (6096 m) [4–7]. With today's industry challenges in meeting ever-increasing production targets and maximizing the full potential from maturing fields, horizontal or complex trajectories are now accepted as standard practice or warranted as a viable design for field development [8]. Figure 2 shows the ratio of the well length vs. the true vertical depth and the reach for wells such as BD-04, M-16, OP-11 and Z-44, which have recently been drilled. When the ratio between the measured depth of wells and the true vertical depths of the wells in Figure 2 is compared with the value mentioned just above, it can be assumed that the aforementioned claim must be revised. Today, we are already distinguishing between low-reach, medium-reach, extended-reach and ultra-extended-reach wells.

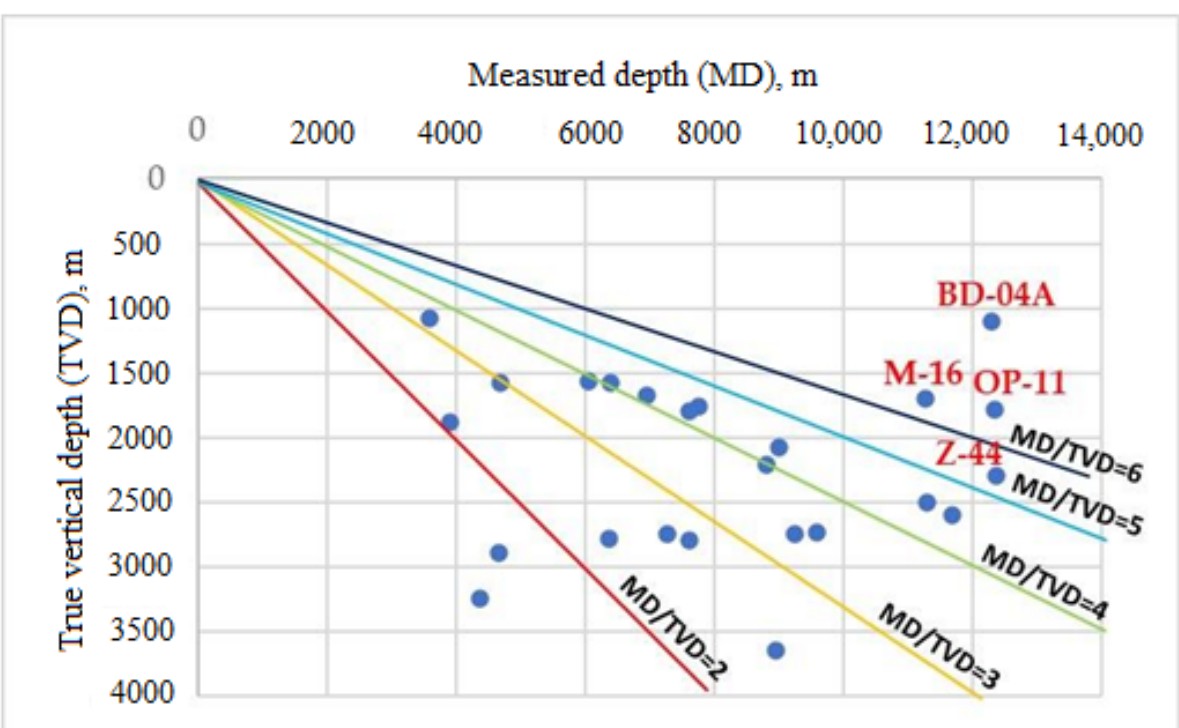

**Figure 2.** The ratio of measured depth vs. the true vertical depth and reach for wells such as BD-04, M-16, OP-11 and Z-44 and other wells presented in Table 2.

With developments and improvements in horizontal drilling and completion technology, there is a constant tendency for operators to drill ever longer lateral wells to increase hydrocarbon production [9]. Drilling a horizontal extended-reach well (ERW) with the shortest time is of great significance for drilling safety, reducing or eliminating reservoir damage, efficiency and economic benefits [10]. This technology enables access to a new energy source that was previously technologically challenging to exploit, such as natural gas hydrate [11]. The open-hole extended-reach limit (the maximum measured depth of the horizontal ERW) is mainly dependent on the annular pressure drop and the fracture pressure of the drilled formation, and the related equivalent circulation density (ECD) of the used drilling fluid. The success of extended-reach well drilling is not only delivering excellent results in maximizing the length of the wellbore and hydrocarbon production, but also in minimizing environmental effects and surface disturbance because fewer wells are needed and surface sites have a smaller footprint [12]. In addition, there is the possibility of drilling more extended-reach/multilateral wells and developing the field from one drilling pad location, significantly reducing the environmental impact of the exploration and production activities. Additionally, the use of special design systems for directional drilling, such as rotation steerable systems (RSSs), results in a more in-gauge hole than drilling with other systems, as well as a precise control of the well trajectory. Thus, this results in lower volumes of drilled cuttings waste and a lower drilling fluid loss [12].

In order to be in line with the project goals, especially during the drilling of the designer well with a complex trajectory, it is necessary to ensure continuously bidirectional communication between the surface location and the equipment on the bottom of the well. The need to transfer the large quantity of information from the bottom hole measuring equipment up to the surface, as well as the need to send information to the steering mechanism in the opposite direction, push forward the development of new data transfer systems. Although mud pulse telemetry is still the main data transfer mechanism in directional drilling operations because of its low data transmission capacity, engineers are trying to further improve it and develop a new data transmission system to meet the new requirements related to the bidirectional transfer of the large amounts of data [13–16]. Since the

maximum capacity for data transfer with mud pulse telemetry has been overcome, further development is in the direction of using wireline drilling tools (now second generation) as well as electromagnetic and acoustic waves. The recent downhole measure-while-drilling technology also allows for the storage of a large quantity of data and subsequent data retrieval once the bottom hole assembly (BHA) is pulled out of the hole. However, wellbore trajectory management during the drilling of the extended-reach wells includes the use of the measure-while-drilling and logging-while-drilling units, which require the application of real-time communication between the BHA and the surface for transferring large amounts of data. Real-time communication is also important for fully automating and optimizing the drilling process, significantly reducing drilling problems and overall costs [17].

Operational challenges associated with drilling extended-reach wells from the drilling fluids' perspective typically include the following: hole cleaning and hydraulic management, equivalent circulating density (ECD) control, narrow mud weight/fracture gradient window, wellbore stability, stuck pipe, lost circulation, torque and drag and barite sag [18,19].

The large number of independent variables influencing cuttings (debris) transport and efficient hole cleaning requires that a computer model is applied to make predictions [20]. Consequently, various computer packages and models have been developed for simulating cuttings transport and predicting hole cleaning in deviated and horizontal wells [21–28]. They allow for an analysis of cuttings transport as a function of operating parameters, wellbore geometry, cuttings characteristics and mud fluid properties. Many researchers have used different machine learning (ML) methods (such as artificial neural networks (ANNs), the Markov rewards process (MRP), random forest regression (RF), linear regression and decision trees) to predict hole cleaning, cuttings transport, local axial velocity, torque and drag, well hydraulics, hydraulic friction factor and pressure losses [29–33].

A special challenge is the design of ERD wells in high-pressure and high-temperature (HTHP) conditions. Cao et al. (2022) established a model to analyze the changing law of the temperature profile inside the production string of an HPHT gas well and recommended changing the well temperature profile by adjusting the production rate or by optimizing the thermal conductivity [34].

This manuscript has three interconnected goals: to give the reader a comprehensive overview of extended-reach technology from the beginning up until the latest technology improvements, single out the main designers and practical problems and solutions in extended-reach technology project implementation and to give detailed analyses of past and current projects and achievements. All of these should be the basis for a comprehensive discussion and answer on the question "Have we reached the technological and economic limit in the implementation of extended-reach technology?"

## 2. Problems with ERD

When drilling an extended-reach well (ERD), various problems can occur, such as high torque and drag values, inadequate hole cleaning, maintaining a smooth wellbore in a challenging well design, differential sticking, steering during stick/slip, bottom hole assembly design, hydraulics and ECD management [35–40].

Issues when drilling an extended-reach well are most often related to poor hole cleaning and high values of torque and drag. This represents a huge problem when drilling an extended-reach well, which is especially pronounced in the deviated and horizontal section of the well as far as torque and drag values are concerned, which can cause limitations in reaching desired depths. Another important issue appears with inappropriate hole cleaning, especially in the deviated and long horizontal sections of the well. Table 1 shows examples of extended-reach wells and the issues that occurred in the well, as well as the solution implemented and obtained results.

**Table 1.** Problems and solutions in extended-reach drilling.

| Source | Well | Field | Location | Mud Type | TVD (m) | MD (m) | Problem | Solution | Results |
|---|---|---|---|---|---|---|---|---|---|
| Mason et al., 1999. [41] | - | Niakuk | Alaska | Low-solids non-dispersed WBM, several proprietary lubricants | 2700–3000 | - | Casing running issues | Casing floatation, centralizers, lubricating agents and the use of drag-reducing roller centralizers were evaluated | Prevent differential sticking, reduce true frictional drag |
| | Gyda A21 | Gyda | Offshore Norway | Oil-based mud | 3600–4300 | - | | | |
| | - | Whych farm | Southern England | - | - | - | | | |
| Trahan et al., 2000. [42] | M-site | Whytch Farm | - | - | - | 11,278 | Installing liner to depth | Floatation used, using only standard Weatherford tools | Floating liner proven to be a constructive idea |
| Cameron et al., 2003. [43] | - | - | Abu Dhabi | Oil-based mud | - | - | Hole cleaning and torque and drag issues | Application of fibrous LCMD sweeps | Sweeps increased cuttings return up to 50%, improvement in ROP, reduced time of completion and torque and drag |
| Elsborg et al., 2005. [44] | Hibernia B-16 36 (OPA1) | Hibernia | Canada | An advanced chemical cleaning system was used to enhance displacement from synthetic-based drilling fluid to water-based completion fluid | 3960.27 | 9356.75 | Hole cleaning, directional and torque and drag issues, tubular design and wear tolerance | Record-length casing strings would have to be deployed, drag reduction techniques | Significant cost savings with 6% NPT |

**Table 1.** *Cont.*

| Source | Well | Field | Location | Mud Type | TVD (m) | MD (m) | Problem | Solution | Results |
|---|---|---|---|---|---|---|---|---|---|
| Walker, 2012. [45] | OP-11 | Odoptu | Sakhalin, Russia | Non-aqueous fluid (NAF) | 1784 | 12,345 | Wellbore instability, shocks and vibrations and high torque | Performance management workflow, casing setting depths were extended to deeper depths, use of liquid lubricants | Established new ERD depth and measured depths records with less than 1% NPT |
| Walker et al., 2009. [46] | Z-12 | Chayvo | Sakhalin, Russia | Non-aqueous fluid (NAF) | 2600 | 11,680 | Torque, hole instability that resulted in difficulties in running liner | Use of lubricants and changes in operating parameters, increasing mud weight to address hole instability. Use of stiffer 7 n. tubing, as well as heavier (additional 5 in. HWDP) | Well completed in 88 days with 9% NPT. Torque friction was 10–12% less than in well Z-11. Lubricants provided 10–12% reduction in torque |
| Al-Ajmi et al., 2013. [47] | RA-492 | Raudhatain | North Kuwait | Oil-based mud (OBM) with water activity in range of 0,75–0,80 and 2–3% proprietary synthetic organic polymer | - | - | Wellbore stability, hole cleaning, highly depleted formations with high porosity and permeability, stuck pipe incidents | Fluid design, combination of calcium carbonate and graphite | Well RA-492 successfully drilled to set record of longest lateral section with 1610,56 m, and production was more than expected compared to offset wells. Conventional oil-based mud was replaced with customized fluid system with a bridging technique in wells RA-493 and RA-499. NPT was reduced by applying a wellbore-straightening package |
| | RA-493 | | | | | - | | | |
| | RA-499 | | | | | - | | | |
| Sonowal et al., 2009. [48] | BD-04A | Al Shaheen | Qatar | Low-solids nondispersive (LSND) water-based mud | ~1100 | 12,289 | Drilling torque friction factor, shocks and downhole vibrations | Lubricants added in 2% to 3%, ECD management, use of 5"x4"drill pipe combination, use of RSS | Record horizontal well 10,902.69 m, 12,289.536 m MDRT and longest along hole departure 11 568.98 m |

Table 1. *Cont*.

| Source | Well | Field | Location | Mud Type | TVD (m) | MD (m) | Problem | Solution | Results |
|---|---|---|---|---|---|---|---|---|---|
| Morrison et al., 2019. [19] | Well A | - | Sakhalin, Russia | Filtered NAF in the open hole, while, in cased hole, displaced to the brine treated with 1% *v/v* lubricant | Less than 2000 | Planned more than 9300 | Torque issues | 1% v/v of Lubricant A added to reduce torque and facilitate the installation of the smart completion | The addition of 1% v/v Lubricant A was observed to reduce the pickup weight by almost 25%. Reduction in torque by more than 50% compared to the brine before the lubricant addition enabled the upper completion to be successfully installed |
| Navas et al., 2016. [49] | - | Upper Zakum | UAE | OBM and SBM | - | Up to 10,668 | Cementing issues. Key challenges: pipe centralization, mud removal, optimizing cement slurry, lost circulation, cement evaluation | Use of new nonwelded single-piece bow spring centralizer. Use of fibers along with high-solid-content trimodel lightweight systems to reduce losses. Fibers and high solid content were used | Operator managed to meet the main zonal isolation in less than 2 years and 15 wells finished with 9 5/8 in. casing cement jobs |
| Dosunmu et al., 2015 [50] | OGG78 | - | Niger delta | - | - | - | Hole cleaning issues causing higher NPT | Real-time cuttings monitoring technique and using other drilling parameters such as torque and drag data to validate it | Reducing non-productive time (NPT) |



### 3. Drilling Fluids for Extended-Reach Wells

In the well-planning phase of ERW construction, special attention should be paid to the choice of drilling fluid. Selected drilling fluids for extended-reach wells should satisfy the same basic functions that are common to all drilling fluids, and they have to provide excellent reservoir protection [51]. When drilling extended-reach wells, the following critical factors should be considered: hole cleaning, torque and drag, borehole stability, equivalent circulating density (ECD) and lost circulation [5,12,18,19,24,51–53].

*3.1. Hole Cleaning*

Drilling fluid has many functions, and one of the primary functions is to carry drilled cuttings to the surface. To achieve that goal, it is necessary to remove them quickly and efficiently. In doing so, it is important to keep in mind that hole cleaning depends on a number of parameters, such as (1) the hole angle of the interval, (2) flow rate/annular velocity, (3) drilling fluid rheology and density, (4) cutting size, shape, density and integrity, (5) rate of penetration (ROP), (6) drill string rotational rate and (7) drill string eccentricity [23,27,54,55]. It is also important to emphasize that effective hole cleaning does not depend on only one drilling parameter but also on a combination of parameters [55]. Bilgesu et al. (2007) divided key factors in drill cuttings transport into three main groups: (a) operational factors, (b) drilling fluid parameters and (3) cuttings parameters. Operational factors include drill pipe rotation, hole inclination, annular eccentricity and the fluid flow rate. The parameters of the drilling fluid refer to its density, rheological parameters and composition. Cutting parameters are their shape, size and type. Only a few of them can be effectively controlled during drilling for hole-cleaning purposes. Poor hole cleaning has led to over 70% lost time in oil and gas drilling operations [27,56].

Unlike vertical wells, in directional, extended-reach wells, there are three cleaning zones that differ from each other according to the hole inclination. These are the I zone (0–30°), II zone (30–60°) and III zone (60–90°) (Figure 3a). As soon as the deviation in the borehole channel exceeds 10°, there is a tendency for cuttings to deposit on the lower walls of that channel. With an increasing inclination, this tendency is more pronounced. However, in practice, the greatest tendency toward the deposition of cuttings was observed in the third zone, at an inclination between 30° and 60° [21].

During the hole cleaning, the cuttings are affected by positive forces upwards (due to the mud velocity, viscosity and density) and negative forces downwards (due to the action of gravity) (Figure 3b).

Cuttings fall or slide through parts of the mud column that do not move (or move slowly) and will fall faster through muds of a lower viscosity than through viscous muds. In order to achieve the satisfactory removal of cuttings from the bottom and bring them to the surface, the annular mud velocity should be slightly higher than the rate of the sliding of the cuttings (cuttings slip velocity, settling velocity) through the mud column toward the bottom of the wellbore. Recommendations for improving the hole cleaning of each zone are presented in Figure 3c.

The annular velocity has the greatest influence on cleaning holes from cuttings in almost vertical (Zone I) and moderately inclined intervals of holes (Zone II), whereas, in extended-reach, high-angle wells (Zone III), it ranks third in importance (Figure 3c). Increasing the flow rate or using a drill pipe with a larger outer diameter (OD) results in a higher annular velocity. In practice, a slight improvement in hole cleaning was observed at an annular rate greater than 60.96 m/min (200 ft/min), but an increase in the equivalent circulating density (ECD) occurred [16]. Unfortunately, increasing the annular velocity increases the flow resistances and hence the ECD, so the flow rate must be balanced to achieve satisfactory cutting transport and to minimize the formation of cutting layers (beds) without creating an excessive ECD.

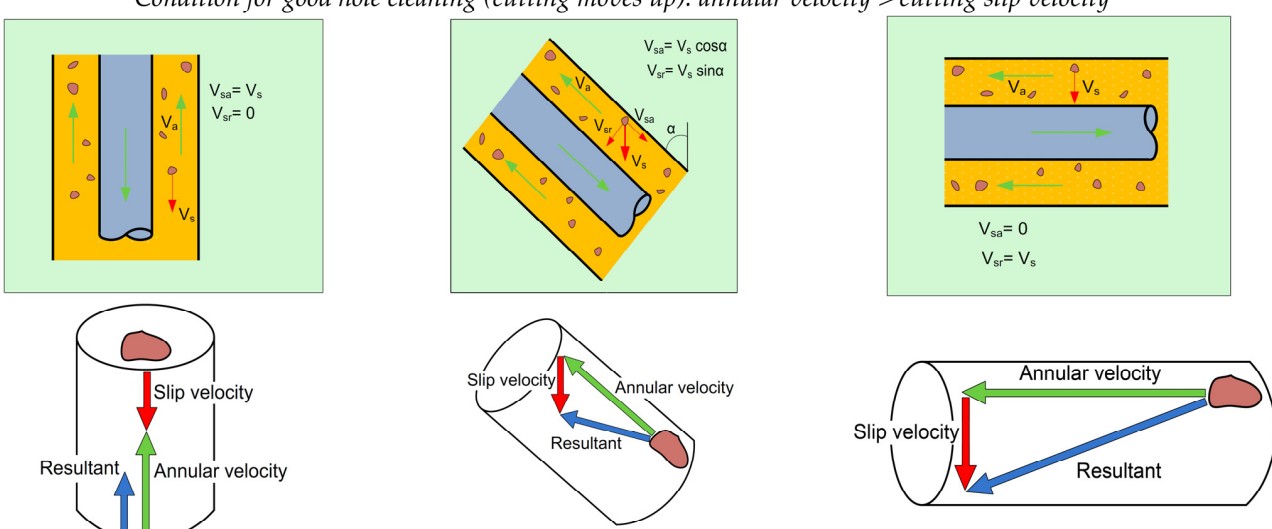

**(a) EXTENDED-REACH WELLS: HOLE-CLEANING ZONES**

**I Zone: 0°–30°**          **II Zone: 30°–60°**          **III zone: 60°–90°**

**(b) INFLUENCE OF GRAVITY AND HOLE INCLINATION ON CUTTING SLIP VELOCITY**

*Condition for good hole cleaning (cutting moves up): annular velocity > cutting slip velocity*

$v_a$ = annular velocity, $v_s$ = cutting slip velocity, $v_{sa}$ = axial slip velocity component, $v_{sr}$ = radial slip velocity component

**(c) RECOMMENDATIONS FOR IMPROVING HOLE CLEANING**

| | | |
|---|---|---|
| 1. Increase flow rate/annular velocity<br>2. Increase rheology (YP or K)<br>3. Decrease ROP | 1. Increase flow rate/annular velocity<br>2. Increase drill string rotational rate<br>3. Flatten rheology profile (6 rpm/LSRV and PV)<br>4. Decrease ROP | 1. Decrease bit cutter size (PDC)/cutting size<br>2. Increase drill string rotational rate<br>3. Increase flow rate/annular velocity<br>4. Flatten rheology profile (6 rpm/LSRV and PV)<br>5. Decrease ROP |

YP = yield point, K = consistency index, ROP = rate of penetration, LSRV = low-shear-rate viscosity, PV = plastic viscosity, PDC = polycrystalline diamond compact

**Figure 3.** Hole cleaning in extended-reach wells (cleaning zones (**a**), influence of gravity and hole inclination (**b**) and recommendations for improving (**c**)).

In Zone I, a laminar flow with a high yield point value and low plastic viscosity (Bingham plastic fluids) or high consistency index value and low flow index value (power-law fluids) will produce a flat viscosity profile and efficiently carry cuttings out of the hole. The yield point may be adjusted with appropriate additives without changing the plastic viscosity significantly. A low plastic viscosity and flat rheological profiles can be achieved if drilled solids are removed from the drilling fluid at the surface using appropriate solid

control equipment to maintain a low drilled solid content in the drilling fluid. If the drilled cuttings are not removed, the plastic viscosity will continue to increase as the drilled cuttings are ground into smaller particles. Large cuttings will separate from the drilling fluid faster than smaller cuttings, but, in high-angle holes, even smaller cuttings may settle and form a cuttings layer [54]. For the efficient transportation of small cuttings during extended-reach drilling, the rotation of the drill pipe in combination with polymer drilling fluid is highly recommended [57]. Pipe rotation can significantly improve hole cleaning, particularly when the drill pipe is eccentric, because effective hole cleaning is not possible with axial flow alone [58].

In zones II and III cuttings, under the action of gravity, they tend to fall through the mud to the lower walls of the hole from which they are only a few millimeters away and form a layer of cuttings. Preventing rock cuttings accumulating on the lower side of the ERD wells is much easier than removing them.

The formation of the cutting bed will be reduced or mitigated if the following are applied: a higher flow rate/annular velocity, higher drill string rotational rate, flatter rheology profile (6 rpm/LSRV and PV) and reduced ROP. The thickness of the cuttings bed will increase from 0.21 m to 0.28 m with an increase in ROP from 15 m/h to 40 m/h [21].

Once a layer of cuttings is formed, an attempt should be made to remove it from the hole. This can be achieved by applying a low-viscosity pill to disturb the cuttings bed, followed by high-density pills (sweeps) of low-viscosity fluid, coupled with pipe rotation. Pipe rotation aids the cuttings removal process, greatly reducing the thickness of the cuttings layer for both low-viscosity and high-viscosity drilling fluids, especially if the pipe is fully eccentric [59]. In the absence of drill pipe rotation, in order to clean cuttings out of a wellbore when drilling is stopped, using water or low-viscosity mud is better than using high-viscosity mud [60].

In horizontal and highly inclined wellbores, drag forces are the main forces that lead to the cuttings bed erosion. Drag forces are higher than the lift forces, which helps to explain the reason why it is more difficult to remove cuttings that have already been embedded in the cuttings bed [61].

Adari et al. (2000) investigated the cuttings bed height as a function of time by using different flow rates (0.76–1.52 $m^3$/min; 200–400 gpm) and four different compositions of the drilling fluid, and concluded that: (a) the erosion of the cuttings bed occurs faster as the drilling fluid flow rate increases, (b) for a given drilling fluid flow rate, a lower cuttings bed height is achieved as the n/K ratio increases, (c) cuttings removal is easier with turbulent flow than with laminar flow, and (d) cuttings accumulation in the wellbore and thus the circulation time required to clean the wellbore from cuttings increases as the inclination of the well increases. The thickness of the cuttings bed has a significant effect on the annular pressure gradient [54,62].

The rotation of the drill pipe not only reduces the solid concentration in annulus but also increases the migration rate of the solid phase, thereby promoting horizontal well cleaning [63]. As the drill string rotates faster, the pipe drags more fluid with it. In deviated and extended-reach wells, this layer of drilling fluid disrupts any debris deposits that have formed around the pipe as it lies on the low side of the well and tends to move it uphole [64]. The drill pipes in ERD are not centered but eccentric, resulting in a high velocity in the upper part of the well and a low velocity in the lower part of the well below the drill pipes. Cuttings are transported better at a high velocity, but the gravity tends to cause the cuttings to fall into the low-velocity area. Moving and rotating the pipe is the only way to transport cuttings into the upper part of the hole above the drill pipes to improve hole cleaning. Creating turbulent flow around the drill pipe can reduce the formation of debris deposits and improve the process of hole cleaning.

Generally, reducing ROP improves hole cleaning. Therefore, the current penetration rate should be controlled so that it is not too large no excessive volume of cuttings is created. In this way, the drilling fluid is given sufficient time to remove the intact cuttings from the

bottom of the well, thus reducing the accumulation of cuttings and avoiding overloading the annular with them.

In order to improve the hole cleaning of the wellbore, there are several methods used in ERD wells, such as optimizing debris removal using real-time annulus pressure measurements during drilling, where real-time annulus pressure monitoring can be used to reduce the risk and optimize the drilling process, especially in extended-reach applications. Field data show how real-time pressure data are used interactively to optimize debris removal and adjust the mud weight. The ability to monitor and subsequently control circulating well pressure allows for narrow acceptance windows between the estimated pore pressure and fracture gradient [65].

The use of downhole mechanical cleaning devices (MCDs) has been introduced into the petroleum industry to alleviate the problem without causing excessive ECD. Recently, hydro-mechanical downhole-cleaning devices such as Hydroclean tools have been developed to increase the efficiency of cuttings transport in directional wells. Hydroclean tools as seen in Figure 4 have helical grooves or blades on their surface to help remove the cuttings layer. Hydro-mechanical hole-cleaning devices improve hole cleaning by creating greater turbulence, bringing debris into suspension and raking the cuttings bed [66].

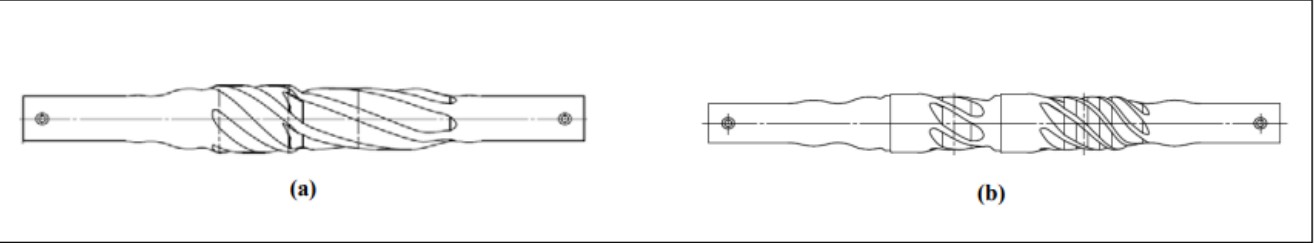

**Figure 4.** Hydroclean tools: (**a**) first-generation tool (G1) and (**b**) second-generation tool (G2) (Reprinted with permission from Ref. [66]. 2023, Society of Petroleum Engineers").

Hole cleaning depends on several factors and, to date, most of the existing models have been applied to solve the problem of hole cleaning. However, the flow rate pre-dicted by these models may not be feasible for practical application in field operations because it produces a pressure that exceeds allowable limits of the pop-up valves on the mud pump. This is a major cause of downtime during well drilling. Dosumnu (2015) developed a model to keep track of the cuttings removal in order to achieve adequate hole cleaning and reduce the non-productive time [50].

The model has the capacity to determine the amount of cuttings collected while drilling in real time, thereby ensuring a reduced non-productive time while ensuring drilling safety as well as the target depth objective. This is possible as a direct quantitative measurement of hole cleaning and hole stability is vital information for reaching these three objectives [50].

Today's needs for larger amounts of hydrocarbons have forced the industry to increase production and thus to improve the way in which oil and gas reservoirs are exploited. The construction of horizontal wells enables the exploitation of oil and gas reservoirs that have been unavailable or economically unprofitable so far, such as the exploitation of reservoirs with a small thickness, the exploitation of reservoirs located under environmentally sensitive areas, etc. Although drilling technology is constantly developing, there are still issues with drilling that occur, and especially with extended-reach wells.

### 3.2. Torque and Drag

The torque or moment is generally the force multiplied by the lever arm. When we talk about torque and drilling, torque is the moment required to rotate the pipe. Torque is used to overcome rotational friction in the well and on the bit. Torque is lost from the rotating string, so less torque is available at the bit for destroying rock. Actual torque and drag values measured in the field are always influenced by other factors. Some of these can

be modeled, while other effects are lumped together into the fudge factor, which we call the friction factor, which is not the same as the friction coefficient as in pure kinetic sliding friction. The combined effect of all of these parameters is what gives the total torque and drag forces. In general, we can separate drag forces that are the result of hole cleaning or inappropriate mud design and drag forces associated with the well path [4].

There are several causes of excessive torque and drag, including tight hole condi-tions, sloughing hole, key seats, differential sticking, cuttings build-up caused by poor hole cleaning and sliding wellbore friction [67].

In extended-reach wells, both in the curved and in the horizontal part of the well, the torsion and drag of drill string are usually larger and there is a tendency for the drill string to lie on the lower wall of the wellbore. Therefore, more complex cleaning conditions are present in ERWs than in vertical and slightly inclined wells.

Torque and drag are important and serious issues for any extended-reach well as they can impose severe limitations on drilling operations. They are affected by many factors, such as the well trajectory design, drilling fluid type, hole size, drill string design and hole cleaning.

The deposition and accumulation of cuttings on the lower wall of the curved part, and especially the horizontal part, of the wellbore contribute to the unfavorable cleaning of the wellbore, with the eccentric position of the drilling tools in relation to the axis of the wellbore.

In addition, the cuttings bed increases the frictional drag and limits the possibility of weight transfer to the bit. This results in an increased possibility of a jammed and stuck pipe, loss of mud, wellbore instability and impossibility or difficulty in well logging and running and/or casing cementing. A reduction or even elimination of these hazards can be achieved by appropriate rheological properties of the mud and a suitable flow regime of the selected mud.

The process of reducing torque and drag is a combination of many different meth-ods. Only one method usually does not solve the problems. In order to reduce torque and drag, several methods may have to be applied at the same time, such as: an optimized well plan, bit selection, drill string design, use of rotary steerable system, use of modular motor, use of non-rotating drillpipe protectors, use of mud additives and proper hole cleaning [68].

The first step in the drilling process design is having an optimized drilling assembly that includes the selection of each particular part of the drill string capable of avoiding any excess in torque and drag. Buster et al. described the performance of High Torque OCTG Premium Connection developed for extended-reach wells with long intervals (>6000 ft) that utilize a thread and coupled design that has an excellent torque capacity [69].

The desire to reduce torque and drag has led to the development of many products that have been successfully applied in practice such as fibrous lost circulation material and glass beads. Numerous lubricants are available on the market that are added to the drilling fluid to reduce the coefficient of friction. According to field experience, a typical lubricant reduces the coefficient of friction by approximately 20%, and a high-performance lubricant by up to 50% [9]. The friction reduction performance of the lubricant for coiled tubing (CT) application in ERWs depends on its concentration and on the presence of polyacrylamide (a viscosifer and fluid friction reducer), salt and sand in the fluid. Laboratory tests have shown that polyacrylamide in a concentration greater than 1% adversely affects the performance of the lubricant while, when increasing the lubricant concentration, the friction coefficient decreases, which is especially pronounced at higher salinity and sand conditions [9].

Besides adding lubricants to the drilling fluid, another solution for additional torque reduction is the use of different mechanical tools. Schamp et al. studied and analyzed the use of mechanical tools to reduce torque in ERD wells. The use of mechanical torque reduction tools consisting of an inner mandrel and an outer sleeve on two wells resulted in significant torque reduction [70].

### 3.3. Equivalent Circulating Density

The equivalent circulating density (ECD) is a combination of the static drilling fluid density and annular pressure loss. The ECD and hole cleaning are inter-related: poor hole cleaning can increase the ECD. The amount of particles present in the drilling fluid increases its density and thus the ECD. In addition, extended-reach wells are characterized by their long horizontal displacement. The annular pressure loss increases when increasing the annulus length, resulting in a continuous increase in the ECD with the measured depth (MD). This is a problem when the formation pressure gradient/fracture gradient window is narrow. The flow rate can be limited to control the annular pressure loss and reduce the ECD, but this can affect the quality of cuttings transport and cause a fluctuation in the ECD. which can cause formation fracturing and a loss of circulation. The presence of cuttings increases the pressure drop due to the reduction in the flow area inside the wellbore. An increase in the annular pressure loss was observed with a high pipe rotation speed in an eccentric annulus [59]. In the absence of cuttings, frictional pressure losses increase as the pipe rotation speed increases, but, in the presence of cuttings, due to a reduction in the stationary area of the cuttings bed, frictional pressure losses may decrease [71].

The open-hole limit of an extended-reach well (the greatest measured depth of the horizontal ERW) mainly depends on the pressure drop in the annulus and the fracture pressure of the drilled formation [72].

### 3.4. Barite Sag

Barite sag occurs when the mud is not circulated for a long time. However, it has been shown that a barite sag can form during circulation and can be thicker than when the flow is static. This process can be accelerated by slow circulating rates, casing running and wire line logging. In extended-reach wells, barite deposition can lead to wellbore mudlosses, mud weight fluctuations, a stuck pipe and wellbore instability. This is una-voidable but can be managed through a combination of good operating procedures and drilling fluid design. A low shear rate viscosity is a critical factor in achieving good hole cleaning and avoiding barite settling [18]. In drilling ERWs, barite sag can be minimized or mitigated by increasing the low shear rate viscosity (LSRV) of the drilling fluid, rotating the drill pipe and using micronized weight materials.

### 3.5. Drilling Fluid Selection

Drilling fluid is extremely important for successful extended-reach drilling (ERD), so it should be chosen especially carefully in order to meet technical, economic and environmental requirements. Historically, oil or synthetic-based muds have tended to be the fluids of choice [18]. ERD drilling fluids are designed to generate a flatter rheological performance to reduce the effect of fluid rheology on the ECD. When selecting drilling fluids for extended-reach wells in areas that are particularly environmentally sensitive, the selection of drilling fluids should take into account technical and environmental criteria re-garding the processing and disposal of cuttings and spent drilling fluid.

Oil-based fluids have been observed in the field to be better at removing cuttings from horizontal wells compared to water-based fluids with similar rheological properties [73,74].

Oil-based muds (drilling fluids) (OBMs) include: a true oil-based mud containing 90–95% diesel oil and 5–10% water emulsified within the oil, an invert emulsion mud containing 60–90% oil and 10–40% water emulsified within the oil, emulsion muds (oil-in-water mud) and synthetic-based mud (SBM) that has a synthesized liquid base (polyalphaolefins (PAO), linear alphaolefins (LAOs), straight internal olefins (IOs), esters, vegetable oils and ethers). Various additives such as viscosifiers, emulsifiers, weighting material and other additives are added to the base fluid to adjust its properties and produce a stable and efficient fluid that will meet the requirements in specific well conditions.

In Western Siberia, oil-based mud was chosen in drilling a 152.4 mm horizontal section on the Samburgskoye field [75]. It provided optimal rheological parameters for shale inhibition, hole stability and lubricity. The implementation of an oil-based drilling fluid

system was justified by the longer lateral section to be drilled. In addition, it helped to improve the RSS steerability and borehole quality [75].

Invert emulsion mud can be formulated with mineral oil or other low-environmental-risk oil substitutes when necessary. In this mud, water and chemicals are used together to control the fluid loss and plastic viscosity. Invert emulsion muds (also known as non-aqueous fluids, NAFs) are the most commonly used oil mud. Invert emulsions generally provide an excellent cuttings integrity, good hole protection and a low coefficient of friction. The latter allows for easier rotation and, in extended-reach drilling, greater flow around the underside side of the drill string. Their use has been a key driver of successful extended-reach drilling and hydrocarbon access [19].

Synthetic-based muds share several advantages with traditional oil-based muds, including improved drilling rates, excellent wellbore stability, reduced torque, good hole cleaning and excellent cuttings integrity. The main advantage of SBMs compared to traditional OBMs is the reduced impact of cuttings and liquid mud on the environment.

By applying increasing environmental restrictions and stricter regulations, the oil industry has been forced to develop water-based inhibited fluid technology, combined with suitable lubricants, that can replace invert emulsion muds.

For extended-reach drilling, the most suitable water-based drilling fluids are those based on potassium, polymer mud with silicates or glycol [12]. These types of drilling fluids are used when shale inhibition is required. Mixed-metal silicates can be used if shale inhibition is not required. Drilling fluids for ERD wells are designed to provide a flatter rheological profile to reduce the effect of the fluid rheology on the equivalent circulating density (ECD) [18].

Burden et al. (2013) examined six water-based muds and one synthetic-based mud. The fluids drilling performance was evaluated on the shale inhibition (bulk hardness, dispersion, accretion and swelling test results) while the environmental evaluation was based on the chloride content, conductance and waste disposal methods. A synthetic-based mud was shown to technically be the strongest, followed by a high-performance amine-modified water-based system that can be combined with techno-economically feasible treatment and disposal options that minimize the environmental impact [76].

To maximize the cuttings removal from the hole, new formulations of water-based muds were developed with the addition of different additives, such as: polymer beads (polyethylene, polypropylene), fibers (monofilament synthetic, polypropylene monofilament, cellulose nanofibers and natural hydrated basil seeds), nanoparticles, bio-based additives and a fuzzy ball [74]. The polymer beads improve the hydrodynamic resistance within the drilling fluid, leading to an increase in the drag coefficient. The fibers are dispersed in sweep fluids to form a stable network structure due to their entanglement. The fiber network prevents cuttings settling by mechanical contact and hydrodynamic interference between cuttings and fibers, and thus improves the drilling fluid carrying capacity. Bio-based additives and organic oils have been proposed to reduce the environmental impact of water-based muds and oil-based muds.

So far, both oil-based mud (OBM) and water-based muds (WBMs) have been used in practice, which can be seen in Tables 1 and 2, as well as the depth and assembly used to drill each of the wells. Tables 2 and 3 summarize available data for 30 extended-reach wells drilled worldwide on 18 production fields.

**Table 2.** The main information about the analyzed extended-reach wells.

| Source | Well | Field | Location | KOP (m) | Measured Depth (MD), m | True Vertical Depth (TVD), m | MD/TVD | Drilling Assembly | Mud Type |
|---|---|---|---|---|---|---|---|---|---|
| Lemons and Craig, 1989. [77] | H-13 | P-0203 block | California, USA | 183 | 3901 | 1877 | 2.08 | N/A | N/A |
| Morgan and Jiang, 1998; Jiang and Nian, 1998. [78,79] | A 14 | N/A | South China Sea | 427 | 9238 | approx 2750 m | 3.36 | kick sub on mud motor | water-based |
| Meader et al., 2000. [80] | M-16 | Wytch Farm | England coast | N/A | 11,278 | approx 1700 m | 6.63 | steerable motor | oil-based |
| Mason et al., 2003. [81] | PN1y | Harding | North Sea | 150 | 6950 | 1676 | 4.15 | - | - |
| | PN1w | | | 150 | 7771 | 1762 | 4.41 | RSS | oil-based |
| | WN1 | | | 150 | 7621 | 1792 | 4.25 | RSS | oil-based |
| | A 16 | Chirag | Caspian Sea | 400 | 7604 | 2800 | 2.72 | RSS | oil-based |
| | A16 T2 | | | 400 | 7280 | 2750 | 2.65 | RSS | oil-based |
| | A17 | | | 200 | 6383 | 2780 | 2.30 | RSS | oil-based |
| | A18 | | | 650 | 9586 | 2730 | 3.51 | RSS | oil-based |
| Schamp et al., 2006. [70] | typical | Chayvo | Sakhalin, Russia | approx 200 | 9100–11,134 | approx 3000 | 3.03–3.71 | RSS | oil-based |
| Sonowal et al., 2009. [48] | BD-04A | Al-Shaheen | Qatar | approx 300 | 12,289 | approx 1100 | 11.17 | RSS | oil-based |
| Mirhaj et al., 2010. [82] | N/A | N/A | North Sea | approx 350 | 5247 | N/A | - | RSS | water-based |
| Walker, 2012. [45] | OP-11 | Odoptu | Sakhalin, Russia | 180 | 12,345 | 1784 | 6.92 | - | - |
| Walker et al., 2009. [46] | Z-12 | Chayvo | Sakhalin, Russia | 200 | 11,680 | 2600 | 4.49 | RSS | oil-based, synthetic-based |
| Gupta et al., 2013. [83] | Z-44 | Chayvo | Sakhalin, Russia | N/A | 12,376 | approx 2300 | 5.38 | RSS | oil-based |
| Okot et al., 2015. [84] | A | Manifa | Saudi Arabia | N/A | 8950 | approx 3650 | 2.45 | RSS | oil-based, synthetic-based |
| Muñoz et al., 2015. [85] | M-1 | N/A | Saudi Arabia | 275 | 11,293 | approx 2500 | 4.52 | - | oil-based |

**Table 2.** *Cont.*

| Source | Well | Field | Location | KOP (m) | Measured Depth (MD), m | True Vertical Depth (TVD), m | MD/TVD | Drilling Assembly | Mud Type |
|---|---|---|---|---|---|---|---|---|---|
| Kretsul et al., 2015. [75] | N/A | Samburgkoye | Western Siberia, Russia | approx 2150 | 4371 | approx 3250 | 1.34 | RSS | oil-based |
| Ahn, 2015. [86] | Control | N/A | N/A | 2118 | 5262 | N/A | - | RSS | oil-based |
| | A | | | 3012 | 6096 | N/A | - | - | - |
| | B | | | 1993 | 4434 | N/A | - | - | - |
| Buster et al., 2016. [69] | typical | Eagle Ford | USA | 1829–3048 | 4877–6096 | 1829–3048 | 2–2.67 | - | - |
| Martinez et al., 2017. [87] | Perla-9 | Perla | Venezuela | 207 | 4660 | 2887 | 1.61 | - | oil-based |
| Golenkin et al., 2020 [88] | 12 | Yury Korchagin | Caspian Sea | N/A | 6061 | 1571 | 3.86 | - | - |
| | 13 | | | N/A | 6390 | 1573 | 4.06 | - | - |
| | 15 | | | N/A | 4684 | 1572 | 2.98 | - | - |
| Vasquez Bautista et al., 2019. [89] | N/A | G | Oman | N/A | approx 3600 | 1078 | 3.34 | - | - |
| Hussain et al., 2021. [90] | A-36 A | Brage | North Sea | approx 350 | 8800 | 2210 | 3.98 | RSS | water-based |
| | A-36 B | | | approx 350 | 9000 | 2079 | 4.33 | RSS | water-based and oil-based |

**Table 3.** The construction of the analyzed extended-reach wells.

| Source | Well | Conductor | | Surface Casing | | Intermediate Casing I | | Intermediate Casing II | | Production Casing/liner | | |
|---|---|---|---|---|---|---|---|---|---|---|---|---|
| | | Diameter, mm (in) | Length, m | Diameter, mm (in) | Length, m | Diameter, mm (in) | Length, m | Diameter, mm (in) | Length, m | Diameter, mm (in) | Length, m | Liner Shoe MD, m |
| Lemons and Craig, 1989. [77] | H-13 | 508 (20) | 133 | 406.4 (16) | 469 | 339.725 (13 3/8) | 1806 | - | - | 244.475 (9 5/8) | 3482 | - |
| Morgan and Jiang, 1998; Jiang and Nian, 1998. [78,79] | A 14 | 609.6 (24) | 205 | 473.075 (18 5/8) | 398 | 339.725 (13 3/8) | 1728 | 244.475 (9 5/8) | 6752 | 177.8 (7) liner | 2578 | 8552 |
| Meader et al., 2000. [80] | M-16 | - | - | 473.075 (18 5/8) | 260 | 339.725 (13 3/8) | 1008 | 244.475 (9 5/8) | 7450 | 177.8 (7) liner | 2921 | 10,210 |
| Mason et al., 2003. [81] | PN1y | - | - | 339.725 (13 3/8) | 2285 | 273.05 × 244.475 (10 3/4 × 9 5/8) | 4663 | - | - | - | - | - |
| | PN1w | | | | 2285 | | 5486 | | | | | |
| | WN1 | | | | 2237 | | 5384 | | | | | |
| | A 16 | | | | 1373 | | 6231 | | | | | |
| | A16 T2 | | | | 1373 | 244.475 (9 5/8) | 5907 | | | | | |
| | A17 | | | | 1377 | | 5006 | | | | | |
| | A18 | | | | 3166 | | 6420 | | | | | |
| Schamp et al., 2006. [70] | typical | - | - | 473.075 (18 5/8) | 800 | 346.075 (13 5/8) | 3300 | 244.475 (9 5/8) | 7800–9600 | 168.275 or 177.8 (6 5/8 or 7) liner | 1300–3200 | 9375–10,900 |
| Sonowal et al., 2009. [48] | BD-04A | 508 (20) | 176 | 339.725 (13 3/8) | 897 | 244.475 (9 5/8) | 1485 | - | - | - | - | - |
| Mirhaj et al., 2010. [82] | N/A | 660.4 (26) | 350 | 339.725 (13 3/8) | - | 244.475 (9 5/8) | - | - | - | 177.8 (7) | 1680 | - |
| Walker, 2012. [45] | OP-11 | - | - | 473.075 (18 5/8) | 800 | 346.075 (13 5/8) | 5254 | - | - | 244.475 (9 5/8) liner | 5652 | 10,758 |

**Table 3.** *Cont.*

| Source | Well | Conductor | | Surface Casing | | Intermediate Casing I | | Intermediate Casing II | | Production Casing/liner | | |
|---|---|---|---|---|---|---|---|---|---|---|---|---|
| | | Diameter, mm (in) | Length, m | Diameter, mm (in) | Length, m | Diameter, mm (in) | Length, m | Diameter, mm (in) | Length, m | Diameter, mm (in) | Length, m | Liner Shoe MD, m |
| Walker et al., 2009. [46] | Z-12 | 762 (30) | 97 | 473.075 (18 5/8) | 801 | 339.725 (13 3/8) | 3313 | - | - | 244.475 (9 5/8) | 8019 | - |
| Gupta et al., 2013. [83] | Z-44 | - | - | 473.075 (18 5/8) | 800 | 346.075 (13 5/8) | 4551 | - | - | 244.475 (9 5/8) liner | 4450 | 8883 |
| Okot et al., 2015. [84] | A | - | - | 473.075 (18 5/8) | 317 | 346.075 (13 5/8) | 1491 | 244.475 (9 5/8) | 3411 | 177.8 (7) liner | 4176 | 7262 |
| Muñoz et al., 2015. [85] | M-1 | - | - | 473.075 (18 5/8) | 275 | 339.725 (13 3/8) | 1850 | 244.475 (9 5/8) | 3375 | 177.8 (7) liner | 5548 | 8608 |
| Kretsul et al., 2015. [75] | N/A | - | - | 339.725 (13 3/8) | 450 | 244.475 (9 5/8) | 1200 | - | - | 177.8 (7) | 3586 | - |
| Ahn, 2015. [86] | Control | - | - | - | - | - | - | - | - | 114.3 (4 1/2) | 5262 | - |
| | B | | | | | | | | | | 4434 | - |
| | A | - | - | - | - | 177.8 (7) | 3297 | - | - | 114,3 (4 1/2) | 3223 | - |
| Buster et al., 2016. [69] | typical | 339.725 (13 3/8)–508 (20) | 45 | 244.475 (9 5/8) | 1524–1829 | 193.675 (7 5/8) | 305–1220 | - | - | 139.7 (5 1/2) | 4877–6096 | - |
| Martinez et al., 2017. [87] | Perla-9 | 762 (30) | 202 | 508 (20) | 642 | 339.725 (13 3/8) | 1893 | 244.475 (9 5/8) liner | 2008 | 127 (5) liner | 4585 | 889 |
| Golenkin et al., 2020 [88] | 12 | - | - | - | - | 273.05 (10 3/4) | 2587 | - | - | 177.8 (7) | 3273 | - |
| | 13 | - | - | - | - | 273.05 (10 3/4) | 2114 | - | - | 177.8 (7) | 3526 | - |
| | 15 | - | - | - | - | 273.05 (10 3/4) | 2303 | - | - | 177.8 (7) | 2464 | - |

**Table 3.** *Cont.*

| Source | Well | Conductor | | Surface Casing | | Intermediate Casing I | | Intermediate Casing II | | Production Casing/liner | | |
|---|---|---|---|---|---|---|---|---|---|---|---|---|
| | | Diameter, mm (in) | Length, m | Diameter, mm (in) | Length, m | Diameter, mm (in) | Length, m | Diameter, mm (in) | Length, m | Diameter, mm (in) | Length, m | Liner Shoe MD, m |
| Vasquez Bautista et al., 2019. [89] | N/A | 473.075 (18 5/8) | 50 | 339.725 (13 3/8) | 229 | 244.475 (9 5/8) | approximately 1250 | - | - | 177.8 (7) liner | - | - |
| Hussain et al., 2021. [90] | A-36 A | 711.2 (28) | 315 | 473.075 (18 5/8) | 1615 | 339.725 (13 3/8) | 1394 | 273.05 (10 3/4) | - | 219.075 (8 5/8) liner | - | - |
| | A-36 B | 711.2 (28) | 315 | 473.075 (18 5/8) | 1615 | 339.725 (13 3/8) | 4574 | 273.05 (10 3/4) liner | - | 219.075 (8 5/8) liner | - | 6935 |

## 4. Analysis of the Presented Case History Data

As can be seen, the drilling and completion of the extended-reach well can be a very demanding engineering task. Despite significant improvements in the directional drilling technique and technology, the elongation of the horizontal displacement on every new extended-reach well creates new engineering challenges. In this section, an analysis of all data from the previous sections will be given. A total of 30 wells from 18 production fields were analyzed (Tables 2 and 3). The main problem with a comprehensive analysis is the lack of data and limitation of available data in the published articles because of company policy about data secrecy.

### 4.1. Well Trajectory

Although the first statement defined extended-reach wells as wells with a ratio between the horizontal reach (departure) and true vertical depth larger than 2, today, those wells can be categorized as horizontal wells. Figure 5 presents the cumulative data for 26 analyzed wells from Table 2, categorized by authors as extended-reach wells. Because there is a lack of data about horizontal reach (departure) for the analyzed wells, the measured depth (MD) and true vertical depth (TVD) were put in a relationship and compared. It is important to emphasize that some authors use exactly this ratio for the classification of extended-reach wells.

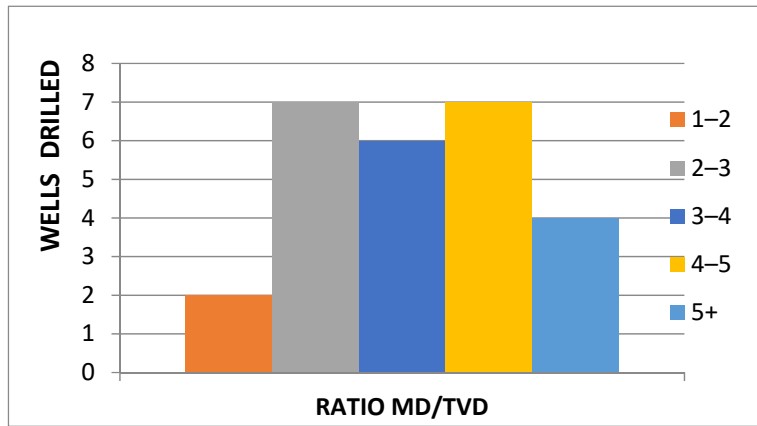

**Figure 5.** The ratio between measured depth (MD) and true vertical depth (TVD) of 26 analyzed wells.

It is evident from the analyzed data shown in Figure 5 that 24 wells have a ratio between the measured depth (MD) and true vertical depth larger than 2, and that this ratio is larger than 5 for 4 of them. According to the last news, Abu Dhabi National Oil Company (ADNOC) this year drilled the longest extended-reach well at its Upper Zakum Concession, with a measured depth of 15,240 m (50,000 ft) [91], overcoming the world record achieved in 2017 in the Chaivo field in the Sea of Okhotsk [92]. Unfortunately, there is no additional information about those two wells, especially detailed information about the well construction, which makes it impossible to compare these boreholes with previously made ones. Furthermore, today, operators use the directional difficulty index (DDI) as the measure for assessing how difficult it is to drill a certain extended-reach well rather than a simple ratio between the horizontal reach (departure) and true vertical depth. Calculating the directional difficulty index information about the total depth of the well, along-hole displacement and total vertical dept is necessary, as well as information about the tortuosity.

The cumulative information about the true vertical depth (TVD) of the analyzed wells from Table 2 are presented in Figure 6.

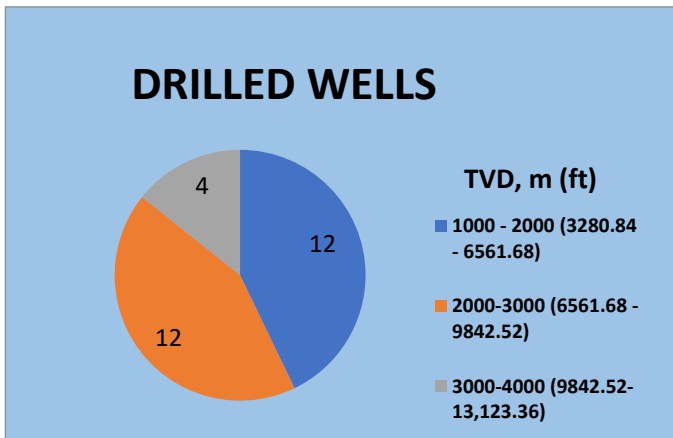

**Figure 6.** True vertical depth of the analyzed wells.

It is evident from the presented data that 12 wells from the analyzed 28 wells with available data have a true vertical depth (TVD) between 1000 m (3280.84 ft) and 2000 m (6561.68 ft), and 12 of them have a TVD between 2000 m (6561.68 ft) and 3000 m (9842.52 ft). Only 4 wells have a TVD larger than 3000 m (9842.52 ft) but no more than 4000 m (13,123.36 ft). These data are in accordance with the main task of drilling an extended-reach well, drilling the well at a relatively shallow depth with a large horizontal departure.

The measured depth (MD) is the only element of the well trajectory available for all 30 analyzed wells (Figure 7).

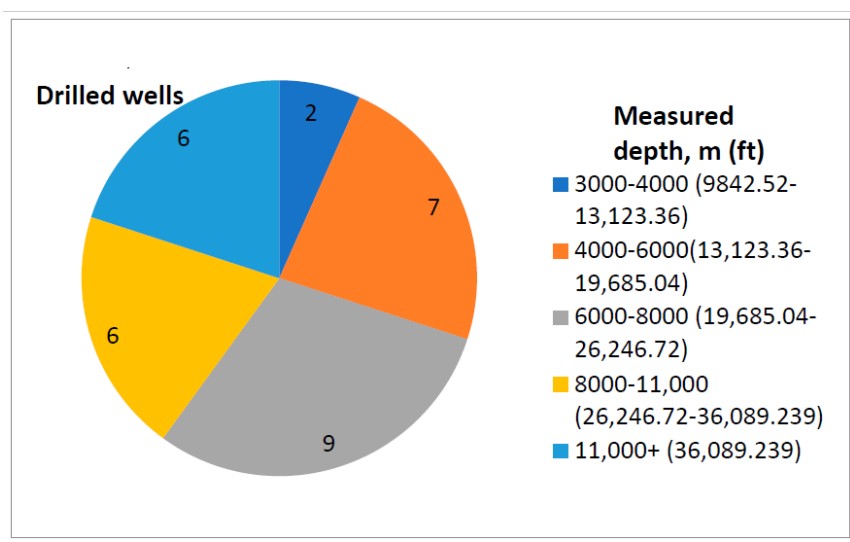

**Figure 7.** Measured depth of 30 analyzed wells.

From the data shown in the picture, it can be concluded that half of the analyzed extended-reach wells have a measured depth (MD) larger than 8000 m (26,246.72 ft), and almost a third of them have a measured depth (MD) larger than 11,000 m (36,089.239 ft). In addition, it is important to note that almost two thirds of the analyzed extended-reach wells have a kick-off point (KOP) at a depth of less than 650 m (2132.55 ft).

### 4.2. Well Construction

The well construction and selection of casing strings are very important from a wellbore stability point of view. The installed casing string should ensure the wellbore stability and integrity during the entire life of a certain well. Although well construction is also important for vertical wells, the design and construction of directional wells and especially extended-reach wells present greater engineering challenges. This is the result of the

complexity of the selected well trajectory, as well as the well tortuosity. Figures 8 and 9 present data about installed conductor strings in the analyzed extended-reach wells. Data are available only for one third of the analyzed wells, and there is no information in the analyzed papers about the reason for the absence of this information. In addition, there is no correlation between the diameter of the conductor string and their setting depth.

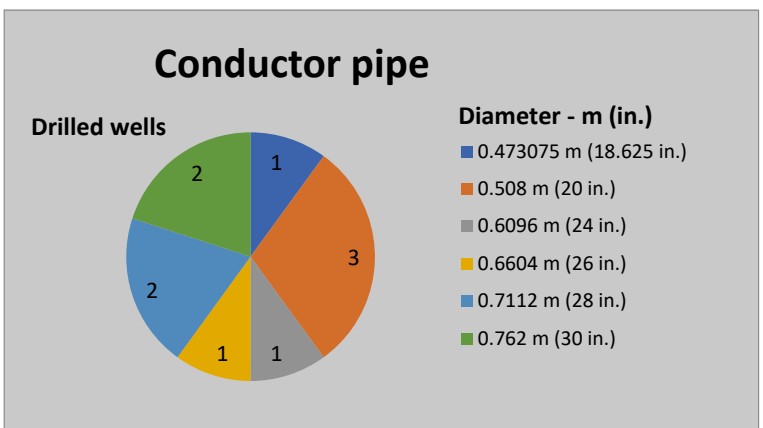

**Figure 8.** Outside diameter of conductor string installed in 10 analyzed wells.

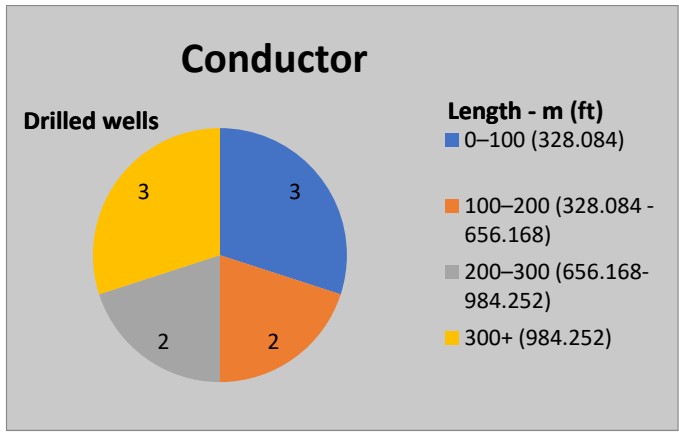

**Figure 9.** Setting depth of conductor strings installed in 10 analyzed wells.

There is some more information about the diameter of intermediate, production or production liner string installed in the analyzed wells (Figures 10–12).

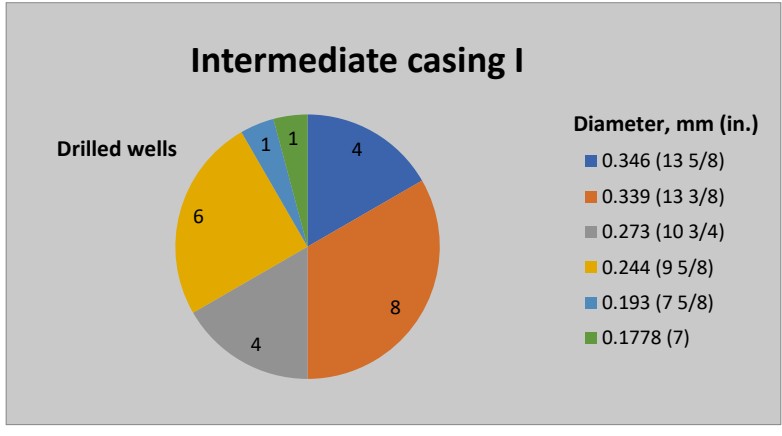

**Figure 10.** Diameter of intermediate casing I in the 24 analyzed wells.

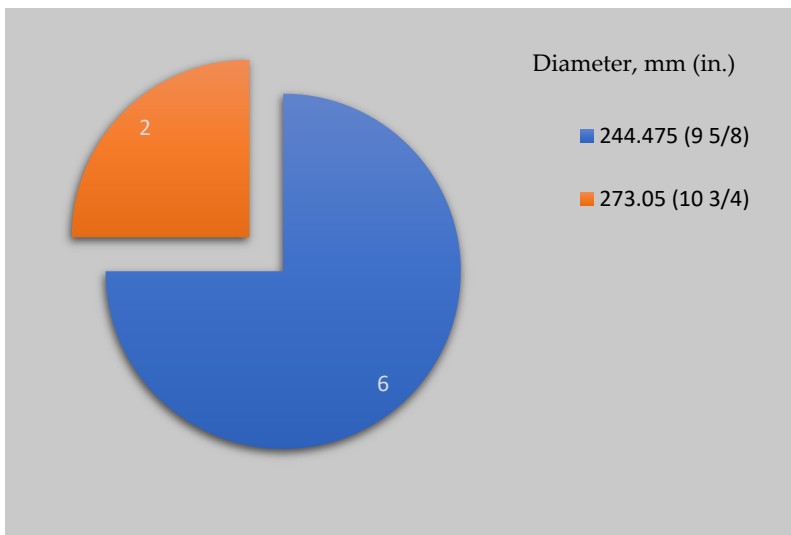

**Figure 11.** Diameter of intermediate casing II in 8 analyzed wells.

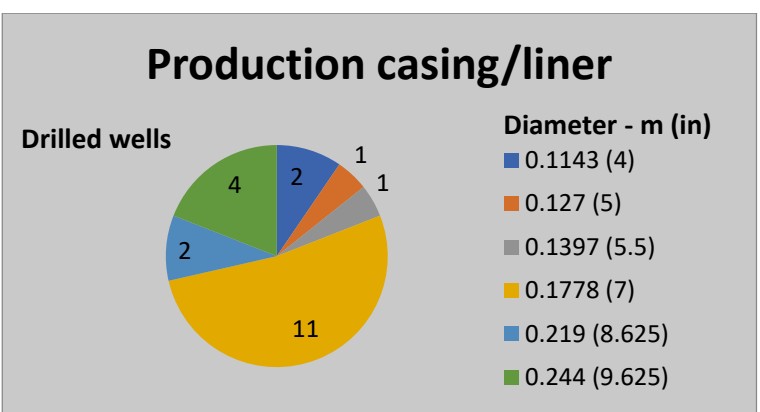

**Figure 12.** Diameter of production casing/production liner in 21 analyzed wells.

Data presented in Table 3 show the technical information for intermediate casing I for 24 wells, where most of the wells have an intermediate casing of 339.725 mm (13 3/8 in.) and 346.075 mm (13 5/8 in.) in diameter. Figure 10 shows the diameter of the intermediate casing I that was installed in the analyzed wells. The widest diameter used is 346.075 mm (13 5/8 in.) and the smallest diameter is 177.8 mm (7 in.).

From the available data given in Table 3 and Figure 11, it can be seen that, from eight wells, six of them had an intermediate casing II with a diameter of 244.475 mm (9 5/8 in.), and the other two wells had an intermediate casing II with a diameter of 273.05 mm (10 3/4 in.). For the rest of the drilled wells from Table 2, there are no available data for the intermediate casing II, and it can be concluded that the majority of analyzed wells were completed without this second intermediate string.

The diameter of the production string or production liner has a range between 114.3 mm (4 1/2 in.) and 244.475 mm (9 5/8 in.), while ap one third of all analyzed extended-reach wells have a production string or production liner that is 177.8 mm (7 in.) in diameter. In addition, one third of the analyzed wells have installed production liners, with different diameters probably due to economic reasons because of the depth of the wells.

### 4.3. Drilling Fluid

According to the data from Table 2, during the drilling of 20 wells, oil-based mud was used in 14 wells, water-based mud was used in 3 wells, oil-based mud and synthetic-based mud were used in 2 wells and water-based and oil-based mud were used in 1 well.

*4.4. Used Directional Drilling Tools*

Table 2 shows that rotary steerable systems (RSSs) were used to drill most of the wells. One well was drilled with a steerable motor and one with a kick sub on mud motor, while for the other wells, there was no information for the drilling assembly. When drilling ERD wells, rotary steerable systems enable the continuous rotation of the drill string from the surface at higher speeds. Different technologies have been used to direct the path into the desired direction: 'push-the-bit', 'point the bit' or their combination (in hybrid RSS). The advantages of RSS technology are: (1) an improved transportation of drilled cuttings to the surface and therefore very efficient hole cleaning and a better hydraulic performance, (2) a reduced torque and drag, (3) an improved rate of penetration (ROP) resulting in a reduced drilling time, (4) a reduced tortuosity of the well and smoother walls of the well, which enables an easier installation of the casing or production strings and better measurements of the formation properties, (5) fewer trips are required because a fixed cutter bit is used and drilling different sections often does not require a change in the BHA design and (6) it is more environmental friendly [12,75].

*4.5. Problems and Solutions during Drilling and Well Completion*

Considering the construction of the extended-reach wells, issues when drilling extended-reach wells are most often related to hole cleaning, high values of torque and drag and wellbore stability, as well as difficulties in running the casing or liner to the predetermined depth. In some cases, operators reported problems with the depleted reservoir, as well as a high-porosity and high-permeability reservoir and, consequently, problems with a stuck pipe and cementation. Most of the perceived problems are successfully solved by the careful selection of the mud type and adjusting their properties (Table 1). Unlike vertical wells, the completion of ERD wells is more complex because difficulties arise when running the casing string or liner to the total depth due to high friction coefficients and an insufficient weight of the upper part of the casing string [12,41]. The solution to the mentioned problem is the use of advanced technology, such as the casing floatation, use of a drag-reducing roller and use of a new non-welded single-piece bow spring centralizer [12,41,49]. The application of casing floatation implies the use of double-floatation collars, filling the upper section of the casing, which is in the vertical section of the well, with mud, and filling the lower casing sections with air [2,12].

**5. Conclusions**

From the first direction well until today, the oil and gas industry has made significant progress in directional drilling techniques and technology. From the first, very modest achievements conditioned by the technology of the time, today, operators and contractors are able to successfully complete the complex design of ultra-extended-reach wells with a measured depth of over 15,240 m (50,000 ft). The drilling of the horizontal and extended-reach well can be observed from economic and ecological points of view. Increasing the contact area between the wellbore and the reservoir also increases the capabilities of the well to produce more hydrocarbons or geothermal water. This means that, at the same time, there is the possibility to cover a large area of the reservoir from one surface location, contributing to a small environmental impact.

According to data presented in the text, it can be concluded that the present directional drilling technique and technology enable drilling directional or extended-reach wells regardless of their directional difficulty index. Recent research in this area has tried to catch up with solutions for the two-way transfer of a large amount of data between the downhole equipment and surface and solve problems related to high values of torque and drag. The solution for the problem related to high values of torque and drag must be economically and environmentally justified in order to replace often-used oil-based mud.

Finally, the further shifting of borders is possible, but there is a question related to the expediency of those projects.

**Author Contributions:** Conceptualization, K.E.S. and B.P.; methodology, K.E.S. and B.P.; validation, N.G.-M.; formal analysis, P.M. and I.M.; investigation, K.E.S.; data curation, N.G.-M. and B.P.; writing—original draft preparation, K.E.S., N.G.-M., P.M., I.M. and B.P.; writing—review and editing, N.G.-M., P.M. and I.M.; supervision, N.G.-M. and B.P. All authors have read and agreed to the published version of the manuscript.

**Funding:** This research received no external funding.

**Institutional Review Board Statement:** Not applicable.

**Informed Consent Statement:** Not applicable.

**Data Availability Statement:** Data available in a publicly accessible repository.

**Conflicts of Interest:** The authors declare no conflict of interest.

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
