# Peer review of "Extended-Reach Drilling (ERD)—The Main Problems and Current Achievements"

_applsci, doi:10.3390/app13074112_

Round 1
Reviewer 1 Report
This is a comprehensive review about ERD well covering theoretical and engineering issues. My concerns are as follows:
1. Machine learning and big data are widely applied in the drilling engineering to predict ROP, gas kick et al. Can it be applied in the hole cleaning and drag analysis for ERD well?
2. Beside lubricant, some shock tools are also developed to reduce the drag. This should be mentioned.
3. According to the latest news, Tarim Oilfield successfully drilling a ultra-deep horizontal well. The total well depth is 9360m and vertical depth is over 8000m. As a result, new challenges are coming for the ERD wells, which are extreme high temperature and high pressure(Cao L, Sun J, Zhang B, Lu N and Xu Y (2022) Sensitivity analysis of the temperature profile changing law in the production string of a high-pressure high-temperature gas well considering the coupling relation among the gas flow friction, gas properties, temperature, and pressure. Front. Phys. 10:1050229). I think it is better to mention the new challenges in the introduction or the analysis.
4. The advantage of rotary drilling tools should be deeply analyzed.
5. Cement and casing running are also different from vertical well. Is there any problem or new technology?
6. The horizontal well are mostly applied in shale gas development. The field data shows that casing damage is easy to find. This paper seems not consider this point.
Author Response
Dear Reviewer,
thank you very much for reviewing our manuscript. Please find the attached file with the answers.
Kind regards

Reviewer 2 Report
The paper is a good review of extended reach drilling. I suggest that this manuscript can be accepted after minor revision.
1. A proof reading by a native English speaker should be conducted to improve both language and organization quality. There are some errors, such as:
(1) The sentence “This paper provides a comprehensive overview of extended reach drilling technology development discusses the main problems and analyses current achievements.” in the abstract is incorrect.
(1) The title of the first picture “Figure 1. Figure 1. Different well construction [2].” is incorrect.
(3)TVD is 3960,266m in the Table1. Please check whether this data is correct.
(4)In the 3.5. section of Drilling Fluid Selection, there is a problem with the text format of line 425.
(2) Extended reach drilling have also investigated in the following literatures. It may help to your work.
[1] Renewable and Sustainable Energy Reviews, 2020, 134: 110388.
[2] SPE Journal, 2018, 23(01): 224-236.
Author Response

(The authors gave the same response as above.)

Reviewer 3 Report
The work in general addresses the challenging issues, the operational summaries, and the future trends associated with the ERD.
The paper has a literature study value and complied the ERD operations drilled around the world. The paper provides a good overview of both well construction and drilling fluid as well the locations.
I believe the work is informative for ERD
Author Response
Dear Reviewer,
thank you very much for reading and reviewing our manuscript.
Kind regards

Round 2
Reviewer 1 Report
The revised paper well answers the comments.